# Anti-*Candida* Activity of Essential Oils from Lamiaceae Plants from the Mediterranean Area and the Middle East

**DOI:** 10.3390/antibiotics9070395

**Published:** 2020-07-09

**Authors:** Giulia Potente, Francesca Bonvicini, Giovanna Angela Gentilomi, Fabiana Antognoni

**Affiliations:** 1Department for Life Quality Studies, University of Bologna, Corso d’Augusto 237, 47921 Rimini, Italy; giulia.potente@unibo.it (G.P.); fabiana.antognoni@unibo.it (F.A.); 2Department of Pharmacy and Biotechnology, University of Bologna, Via Massarenti 9, 40138 Bologna, Italy; giovanna.gentilomi@unibo.it

**Keywords:** essential oils, Lamiaceae plants, *Candida albicans*, *Candida* non-*albicans*, cytotoxicity, anti-virulence factors

## Abstract

Extensive documentation is available on plant essential oils as a potential source of antimicrobials, including natural drugs against *Candida* spp. Yeasts of the genus *Candida* are responsible for various clinical manifestations, from mucocutaneous overgrowth to bloodstream infections, whose incidence and mortality rates are increasing because of the expanding population of immunocompromised patients. In the last decade, although *C. albicans* is still regarded as the most common species, epidemiological data reveal that the global distribution of *Candida* spp. has changed, and non-*albicans* species of *Candida* are being increasingly isolated worldwide. The present study aimed to review the anti-*Candida* activity of essential oils collected from 100 species of the Lamiaceae family growing in the Mediterranean area and the Middle East. An overview is given on the most promising essential oils and constituents inhibiting *Candida* spp. growth, with a particular focus for those natural products able to reduce the expression of virulence factors, such as yeast-hyphal transition and biofilm formation. Based on current knowledge on members of the Lamiaceae family, future recommendations to strengthen the value of these essential oils as antimicrobial agents include pathogen selection, with an extension towards the new emerging *Candida* spp. and toxicological screening, as it cannot be taken for granted that plant-derived products are void of potential toxic and/or carcinogenic properties.

## 1. Introduction

*Candida* spp. are the most important cause of opportunistic mycoses worldwide. Yeasts of the genus *Candida* are associated with different clinical manifestations ranging from superficial infections, those involving the skin and mucosal surfaces, to systemic and potentially life-threatening diseases in otherwise healthy individuals. Overall, *Candida* spp. are one of the primary causes of catheter-associated bloodstream infections in intensive care units of U.S. and European hospitals, and the fourth most common cause of nosocomial bloodstream infection in the USA [1,2]. These severe infections are associated with high mortality rates that are difficult to ascertain, as many patients who acquire candidemia have an underlying medical condition. However, data from population-based surveillance studies report mortality rates ranging from 29% in the USA to 72% in Brazil [3].

Although more than 100 species of *Candida* have been described, the most challenging infections are caused by *C. albicans, C. glabrata, C. parapsilosis, C. tropicalis*, and *C. krusei. C. albicans* is still regarded as the most common species, even though during the last 20 years a progressive shift in the etiology of candidiasis from *C. albicans* to other species has been observed. Currently, approximately half of the cases of candidiasis are caused by non-*albicans* species, such as *C. glabrata*, *C. parapsilosis*, *C. tropicalis*, *C. krusei*, *C. guilliermondii*, and *C. dubliniensis*. The overall species distribution of *Candida* spp. is dependent upon geographic location and patient population, however, the steadily increasing isolation rate of *C. glabrata* in the USA, Canada, northern European countries, and Australia, *C. parapsilosis* in Latin America, southern European countries, and Africa, and *C. tropicalis* in Asia is well documented [3,4].

Among *Candida* spp, antifungal resistance is observed towards the three main antifungal drug classes: the azoles, the echinocandins, and the polyens. While primary resistance to fluconazole is found naturally in *C. krusei*, secondary resistance mainly concerns *C. albicans C. glabrata*, *C. parapsilosis*, *C. tropicalis*, *C. krusei* to echinocandins and *C. albicans, C. glabrata*, and *C. tropicalis* to amphotericin B. In addition, coevolution of azole and echinocandin multidrug resistance in *C. glabrata* has substantially increased along the last decade [5,6].

Current epidemiology of candidiasis has been further modified by the recent identification of *C. auris*, first discovered in Japan in 2009. By 2018, cases of *C. auris* infections had become widespread across the globe. The infection is classified as ‘urgent threat’, as the yeast is multidrug resistant, causes high mortality (17–72%), spreads easily in hospital settings, and is difficult to identify [7,8].

Despite the latest developments of diagnostic tools and therapeutic options, candidiasis still remains difficult to treat regardless of its etiology, due to the characteristics of *Candida* spp., such as resistance to antifungal drugs, expression of virulence factors, and the ability to form biofilms, which is nowaday considered the dominant form of pathogens in natural infections [9,10]. Each *Candida* species exhibits differences in terms of biofilm formation; architectures, cellular morphologies (yeast cells, hyphae, and pseudohyphae), and extracellular matrix composition are not only species-specific but, in some cases, also strain-specific. Fighting these structures turns out to be even more difficult from a clinical point of view [10,11].

In order to cope with this scenario, the international research community is engaged in discovering new, potent, and promising compounds to be used alone and/or in combination with commercially available antifungal drugs. In addition, an alternative approach to combat *Candida* infections has recently gained attention, that is, to target functions crucial for the pathogen’s virulence [12].

Plants provide unlimited opportunities for isolation of new compounds because of the unmatched availability of chemical diversity and, therefore, many studies describe the potential antimicrobial properties of plant extracts, essential oils (EOs), and pure secondary metabolites. Faced with the ever-growing amount of scientific data on anti-*Candida* inhibitors, there is a need to critically review current knowledge on natural products.

This review aims to discuss the scientific documentation on the anti-*Candida* properties of EOs from plants of the Lamiaceae family growing in the Mediterranean area and the Middle East. Current knowledge regarding changes in the epidemiological features of *Candida* spp. and the emergence of pathogens with decreased antifungal susceptibilities are discussed in order to provide an up-to-date overview of antimicrobial research.

## 2. Essential Oils from the Lamiaceae Family

The Lamiaceae family, commonly known as the mint family, is a large family of flowering plants, with 236 genera and more than 7500 species with an almost worldwide distribution, but with the majority of species inhabiting Eurasia and Africa [13]. It is one of the plant families well known for its fragrant species, many of which are of highly valued in the cosmetic, perfumery, food, and pharmaceutical industries [14]. The fragrance of Lamiaceae plants is due to EOs, mixtures of specialized plant metabolites that are produced and accumulated within specialized structures, such as glandular hairs (trichomes), oil ducts, or secretory pockets, localized in different plant organs. Leaves and flowers represent the main storage organs for EOs, but other parts of the plant, including seeds, fruits, rhizomes, and even bark can also contain them. From these organs, EOs can be extracted by a variety of methods, including steam-distillation, hydro-distillation, dry-distillation, solvent and supercritical fluid extraction [15,16], even though only steam- and water-distillation can be used to obtain an EO as defined by the ISO in document ISO 9235.2.

From a chemical point of view, EOs are complex mixtures of organic volatile compounds belonging to different classes. The chemical complexity of these oils is remarkable, and up to 300 different compounds can be present in an EO [17]. Among them, terpenes represent the predominant compounds, but phenylpropanoids also occur. Within the class of terpenes, monoterpenes and sesquiterpenes are the most abundant constituents, although their derivatives, including alcohols, aldehydes, ketones, esters, and phenols, are also common in variable proportions. Synthesis of terpenes and phenylpropanoids occurs in the plant cell through different metabolic pathways; monoterpenes and sesquiterpenes are synthesized through the non-mevalonate and the mevalonate pathways, respectively, while phenylpropanoids arise from phenylalanine and tyrosine that, in turn, originate from the shikimate pathway.

Monoterpenes of EOs comprise both aromatic and acyclic structures, with the former representing the largest group of naturally occurring monoterpenes. Among them, common compounds include α-terpinene, β-terpinene, γ-terpinene, limonene, α-pinene, β-pinene, *p*-cymene, and its hydroxylated derivatives tymol and carvacrol, as well as the notable pulegone and piperitone. Some acyclic monoterpenes are also important constituents of EOs, including linalool, geraniol, and citronellol. As concerns sesquiterpenes, they show a wide structural diversity, with linear, branched, or cyclic structures [18,19]. Examples of compounds belonging to the latter group include the azulenes, that are responsible for the blue color of some EOs, α-bisabolene and its oxygenated derivatives, α- and β-bisabolol, and caryophyllene. The latter is present, as β-caryophyllene, in many EOs and, in some cases, represents the major component [20]. Phenylpropanoids occur less frequently in EOs and, when present, are usually less abundant than terpenes. Examples of important phenylpropanoids include cinnamaldehyde, myristicin, dillapiole, anethole, chavicol, eugenol and their methylated derivatives [21].

The plant species herein reviewed were sourced from Wos of Science and PubMed databases by using “*Candida*”, “essential oil”, “Lamiaceae” as key words, alone and as combinations. Only papers published from 2010 to March 2020 were analyzed. Plant species growing in the Mediterranean and Middle East was included as a specific criterion to filter reports; a total of 100 plants were identified and are, therefore, considered in this review. Italy, Iran, and Turkey were the most representative countries (17.0, 13.8, and 9.6%, respectively), followed by Portugal (8.5%), and Algeria (7.5%), while other countries of these areas had a relative distribution in the range 1.0 to 7.0% (Figure 1).

The 100 most frequently investigated plants belong to 24 genera, among which are *Thymus*, *Mentha*, *Salvia*, and *Origanum*, with percentage distributions of 22.6% 11.3%, 9.4%, and 8.5%, respectively. *Calamintha, Stachys*, and *Lavandula* are also well represented (5.7%). The compositions of the EOs are listed in Appendix A; scientific plant names were checked using The Plant List database [22] and chemical components were unified according to PubChem [23].

## 3. Selection of *Candida* Spp.

The EOs from Lamiaceae plants that have been identified as active towards *Candida* spp. were assayed in vitro against reference strains and/or clinical isolates (Table 1). Reference strains were tested in 76.4% of the papers and were obtained from four organizations: American Type Culture Collection (ATCC), Agricultural Research Service Culture Collection (NRRL—Northern Regional Research Laboratory), CBS-KNAW Culture Collection (CBS), and Moroccan Coordinated Collections of Microorganisms (CCMM). Testing reference strains should be preferred in primary screening studies as these microorganisms are well-characterized and widely used, thus allowing for a direct comparison of literature data. Clinical strains from biological specimens were assayed as unique targeted pathogens in 21.3% of the papers and the majority of these isolates were not defined for their susceptibility to relevant antimicrobial agents. This information should be included in the research studies for a proper description of the tested samples and may represent an added value to determine the effectiveness of natural compounds towards pathogens circulating in the population.

*C. albicans* constituted the primary target for the assessment of the anti-*Candida* activity of the EOs, both as reference strain and as clinical isolate. Testing a single *Candida* spp. is not a drawback, but a feature to be considered in relation to epidemiological changes occurring in *C. albicans*. In addition, the recent emergence of novel, multiresistant species, such as *C. auris*, amplifies the call for testing this pathogen. Thus, it is desirable that researchers improve their antimicrobial evaluations on natural products considering current clinical needs.

## 4. Anti-*Candida* Activity

Antimicrobial activity of natural extracts and pure compounds can be evaluated by measuring the growth response of microorganisms to samples that are placed in contact with them. Different methodologies for in vitro tests are available and roughly classified in two main groups: diffusion assays (agar disk, agar well, agar plug diffusion methods), and dilution assays (agar dilution and broth micro-, macro-dilution methods) [113]. As these methodologies incorporate viable cells whose growth can be influenced by unpredictable factors, a carefully standardization of the techniques is of utmost importance. Some methods were subjected to standardization by the CLSI (Clinical and Laboratory Standards Institute) and EUCAST (European Committee on Antimicrobial Susceptibility Testing), marking the major remarkable steps on the procedures.

The two most common methods used to investigate anti-*Candida* activity of EOs collected in the Mediterranean area and the Middle East were agar disk diffusion and Minimum Inhibitory Concentration (MIC) assays. As is desirable, the test of choice resulted in MIC measurements and 95.5% of EOs were assayed by broth dilution tests. Indeed, dilution methods in broth are the gold standard for antimicrobial susceptibility testing, and are mainly used to establish the in vitro activity of new antifungal agents, such as substances of biological, semi-synthetic, or synthetic origin that inhibit the growth of fungi or are lethal to them [114]. Agar-based assays are practical tools, due to their simplicity and capacity to analyze a large number of samples, however they are qualitative tests where diffusion plays an important role in determining the size of growth inhibition zones. Problems may arise especially when investigating EOs, as they are lipophilic and volatile; thus, they do not easily diffuse through agar and their evaporation may impact on the outcome of the assay. In addition, papers herein reviewed report data obtained in experiments where the EO samples were placed on different reservoirs in terms of types and dimensions, making comparisons between results on potency unreliable. As an example, the anti-*Candida* activity of *Thymus capitatus* (L.), Hoffmanns. and Link [78], was tested in an agar well (Ø 10 mm) bored into the culture medium, while *Mentha cervina* L., *Ocimum basilicum* L., and *Origanum vulgare* L. [59] were spotted on sterile paper discs Ø 9 mm), while standard guidelines indicate sterile paper discs of Ø 6 mm.

Results concerning the anti-*Candida* activities of the selected EOs are reported in Appendix A and are expressed as MIC values obtained by broth micro- and macro-dilution tests. This specific criterion was selected in order to compare a more homogeneous dataset.

Activities extend over a wide range of values (0.39–12,480 µg/mL; 0.125–40 µL/mL) regardless of the EO and the targeted *Candida* spp. Classification of the EO’s potency as strong, moderate, or weak, is a very difficult procedure and even authors of the selected publications point out significant discrepancies in anti-*Candida* properties. For example, *Origanum ehrenbergii* Boiss. and *O. syriacum* L. with MICs of 800 µg/mL are considered inactive by Al Hafi et al. [28], while *Micromeria inodora* (Desf.) Benth. with a MIC of 1000 µg/mL is regarded as a moderate inhibitor by Benomari and coworkers [38]. In many papers, the anti-*Candida* inhibitory potential is evaluated with reference to antifungal commercial drugs, e.g., fluconazole and amphotericin B; however, the activity of pure compounds cannot be compared with complex and structurally diverse natural mixtures, and their MIC values have to be considered as positive controls.

In an attempt to categorize EO potencies, the overall MIC values obtained for *C. albicans* reference strains are plotted in Figure 2. The frequency of distribution of MIC values indicate that more than half of the tested EOs is active in the 100–1000 µg/mL and 0.31–2 µL/mL range, indicating 1000 µg/mL and 2 µL/mL as thresholds to define an EO as an antifungal inhibitor. Table 2 reports MIC values of the EOs displaying inhibitory activity towards *Candida* spp. reference strains.

In terms of potency against *C. albicans* strains, some EOs showed a strong activity, with MIC levels <100 µg/mL or <0.3 µL/mL. Even though the concentration of 100 µg/mL has been adopted as a general endpoint criterion for plant-derived mixtures in all anti-infective bioassays [115], less than 10% of selected species meets that criterion, while most species produce an EO with a lower level of activity. This can be related to the intrinsic chemical nature of EOs and, in particular, to the low solubility of most of their components in aqueous media. Plants producing an EO with MIC values on *C. albicans* <1000 µg/mL or <2 µL/mL include several species of *Thymus, Coridothymus, Origanum, Mentha, Calamintha, Satureja, Salvia, Lavandula, Plectranthus,* and *Stachys.* Within this group, it is worth noting that some EOs were active at concentrations <100 µg/mL, such as *Mentha mozaffarianii* [102], *Mentha suaveolens* Ehrh [51], *Salvia mirzayanii* Rech.f. and Esfand [56], *Stachys spruneri* Boiss. [71], *Thymus willdenowii* Boiss [84], *Plectranthus barbatus* Andrews, and *Plectranthus caninus* Roth [55]. When considering MIC values expressed as µL/mL, the two most active species were *Nepeta cataria* L. [108] and *Zataria multiflora* Boiss [82], with MIC values <0.3 ml/L.

The most frequently represented chemical constituents of EOs endowed with anti-*Candida* activity belong to the group of monoterpenes, and include *p*-cymene (40 plants), linalool (35 plants), γ-terpinene (33 plants), carvacrol (31 plants), 1-8-cineole (30 plants), α-pinene (28 plants), and thymol (27 plants). The sesquiterpene β-caryophyllene is present as a constituent in 15 out of 100 plants. 

Given the high chemical complexity of EOs and considering that the biological effect are often the result of a synergistic interaction occurring among the various components of the mixture, it is difficult to univocally identify the most active components. Based on the evaluation of the MIC values on *C. albicans* reference strains, it appears that some monoterpenes and derivatives were present as major constituents in EOs endowed with a powerful antifungal activity. Among these, terpinyl-acetate, α-terpineol, β-linalool, and γ-terpinene confer a strong anti-*Candida* activity to the EO when present as major components, as in the case of *Salvia mirzayanii* Rech.f. and Esfand [56], *Plectranthus caninus* Roth [55], and *Thymus willdenowii* Boiss. On the other hand, compounds making up a larger proportion of the EO are not necessarily responsible for the activity, and this makes it even more difficult to establish a correlation between chemical composition and biological efficacy. 

The antifungal activity of single acyclic and cyclic monoterpenes has been deeply investigated [116,117,118]; α-terpineol, terpinen-4-ol, 1-8-cineol, and β-linalool were reported to show the most rapid killing activity, while γ-terpinene, α-terpinene, terpinolene, and *p*-cymene, showed a slower, although still significant, antifungal activity. This suggests that the alcohol moiety, more than the cyclic or acyclic structure itself, is important for a rapid inhibitory effect on fungal growth, and this has been related to the higher water solubility of alcohols in both aqueous media and microbial membranes [118,119]. Indeed, the antifungal effect of linalool has been demonstrated to be the result of the combined action on membrane integrity and cell cycle arrest [120,121]; moreover, an inhibitory action on biofilm formation, occurring through an impairment in filament production, has been documented [122]. The alteration of cell permeability caused by the insertion of small molecules of monoterpenes between the fatty acid chains of membrane phospholipids seems to be a common effect [123,124]. A high inhibitory activity was related to the presence of an aromatic ring in the monoterpene molecule [125], and thymol and carvacrol have been the subject of several investigations aimed at unravelling their mechanism of action. This involves the inhibition of ergosterol biosynthesis, with disruption of membrane integrity [126]. Recent papers reported that carvacrol exerts its potent antifungal activity by altering the integrity of endoplasmic reticulum, thus impairing the normal protein folding capacity of *C. albicans* cells.

## 5. Anti-Virulence Activity

The pathogenicity of *Candida* spp. is attributed to several virulence factors including yeast-hyphal transition, adherence to surfaces, production of hydrolytic enzymes (proteases, lipases, hemolysins), signal transduction pathway, and biofilm formation. In human infections, the formation of a germ tube and mycelium is an essential process for host tissue invasion, damage to mucosal epithelia, escape from host immune cells, and blood dissemination; additionally, filamentation is pivotal for robust biofilm development. Thus, natural products affecting both morphogenetic transitions between yeast and filamentous morphologies and biofilm formation represent alternative candidates for the treatment of candidiasis; currently, strategies targeting the expression of virulence factors are a promising approach for the treatment of infectious diseases, since these drugs can reduce the risk of resistance development in host infections [9,12].

Among the EOs investigated in the publications reviewed herein, the anti-*Candida* activity through inhibition of virulence factors was evaluated for 13 of them. In almost all papers, the targeted yeast was *C. albicans* reference strains and the EOs were tested in vitro for their effect on the hyphal development process and/or for their anti-biofilm activity. Several approaches are available to determine these properties [127]; the most common test to study yeast-to-hyphae transition is by culturing *C. albicans* in medium supplemented with serum at 10%, followed by light microscope observation, while biofilm production, inhibition, and eradication can be assessed by crystal violet staining of the biofilm mass obtained on tissue-culture plates. A representative sketch describing the effects of an anti-virulence agent on a *C. albicans* culture is depicted in Figure 3.

Table 3 reports the data of the anti-virulence effects for 13 EOs; the yeast-to-hyphae transition was affected at sub-inhibitory concentrations for species of *Lavandula*, *Mentha, Origanum*, *Thymus*, and *Ziziphora tenuior* L., the latter species being also capable of decreasing biofilm biomass, together with some other species of *Satureja*, *Thymbra*, and *Thymus.* These findings add relevant information to the anti-*Candida* activities of these EOs, considering that biofilms display innate resistance to multiple drug classes and are capable of withstanding antifungal concentrations 1000-fold higher than those that inhibit planktonic cells [128].

Thymol, carvacrol, and 1,8-cineole were reported to be effective in inhibiting germ tube formation, an initial stage in the transition from yeast to hyphae, with an efficacy comparable to that of the “quorum sensing” molecule farnesol [129]. In another study carried out by Boni et al. [130], pulegone and carvone were found to be highly active molecules in blocking the progress of biofilm formation and in damaging mature biofilms at low concentrations. These findings can account for the anti-virulence activity of EOs shown in Table 3.

## 6. Synergistic Interaction with Commercial Antifungal Drugs

Drug combination approaches to tackle *Candida* infections are of great interest; indeed, much effort is put in the search for molecules leading to synergistic interactions with commercial drugs to achieve lower effective doses of drugs and to restore antifungal activities when drug resistance arises [131,132]. However, the majority of these combination studies were performed in vitro, thus lacking evidence for the in vivo and clinical effects, which limits the possible development of anti-candidiasis drugs with some general concerns about potency and potential toxicity.

In particular, studies examining the effects of EOs and conventional drugs are under-represented in antifungal drug discovery research and more investigations are required. Indeed, among the papers herein reviewed, synergistic interaction was described only for three EOs from Lamiaceae plants of the Mediterranean area and Middle East: *Thymus vulgaris*, *Coridothymus capitatus*, and *Mentha x piperita*. Synergy was demonstrated for *T. vulgaris* in combination with amphotericin B, even though a non standardized assay was applied [60]. The most common method aimed at evaluating the interaction between molecules is the checkerboard assay and the type of interaction is usually described by the fractional inhibitory concentration (FIC) index, a value that takes into account the potency of the combination of molecules in comparison to their individual activities [133]. *Coridothymus capitatus* differentially interacted with itraconazole, depending on the targeted *Candida* spp.; of clinical relevance is the synergistic effect of the EO with itraconazole against *C. krusei,* which is intrinsically resistant to fluconazole [74]. The combinatorial interaction of *Mentha x piperita* with commercial antifungal drugs (several azoles and amphotericin B) was investigated on different reference and clinical strains of *Candida* spp. [92,103]. Overall, results point to the synergy of the EO with amphotericin B, fluconazole, and miconazole towards reference strains, and with itraconazole towards clinical isolates of *C. albicans*, *C. glabrata*, and *C. krusei* with different azole-sensitivity. Remarkably, when investigations were carried out for menthol and menthone, the two main components of the EO, only an additive effect was recorded, confirming that the activity of an EO is due to the whole phytocomplex that, in fact, better expresses its effectiveness.

All synergistic combinations can be ascribed to the cell wall or cell membrane disruption mediated by the EO, which enhances penetration of antifungal drugs across the cell barriers [130,134].

## 7. Cytotoxicity

Assessment of the toxicological profile of natural compounds is a very important step in a drug discovery perspective. Cell viability and proliferation assays on mammalian cells should always be included in the investigations, since they allow to discriminate between a specific antimicrobial activity and a non-specific cytotoxicity. Many cell types and bioassays can be used for this purpose; natural compounds, at least at bioactive concentrations, should be evaluated on normal and/or malignant cells and a variety of methodologies are available, such as enzyme activity, cell membrane permeability, cell adherence, ATP production, co-enzyme production, and nucleotide uptake activity [135].

Unexpectedly, cytotoxicity was evaluated in only 20.2% of the investigated papers; cell viability was ascertained primarily by using the standardized MTT assay, an enzyme-based method relying on cellular dehydrogenase activity, and in more than one cell line. *Artemia* spp. (brine shrimps) have also been used as a biological model in three research papers [39,40,69], however, they are only suitable for ecotoxicity studies [136].

Table 4 reports the MIC values of 20 EOs tested against *Candida* spp. reference strains and data concerning their effects on mammalian cells, expressed either as percentage of cell viability compared to control cells, at a defined concentration, or as the IC_50_ value, the concentration at which viability was reduced by 50%.

Fungi and mammalian cells share similar cellular and biochemical pathways, thus some degree of toxicity for EOs is not surprising. Cytotoxicity against cancer cell lines could be considered an added value for EOs in the framework of anti-tumor drug development [33]; on the contrary, reduced cell viability on non malignant cells should be carefully considered when EOs are investigated as the potential source of antimicrobial agents. As reported in Table 4, only seven EOs were assayed on normal cells, thus indicating the need for a more careful choice of cells to be used in the experiments. The selectivity index (SI), the ratio between cytotoxicity and antimicrobial activity, is a widely accepted parameter used to express the in vitro efficacy of a test sample and to rule out a general effect on eukaryotic cells; to obtain the SI value, inhibitory activity towards a specific target has to be expressed as IC_50_ rather than MIC. Considering that the potency of EOs is reported as MIC, the SI cannot be measured and conclusive statements on the selective activity of EOs and on their safety is hazardous. Indeed, several EOs listed in Table 4 strongly interfere with cell proliferation of the tested cell lines with IC_50_ values <30 µg/mL, a threshold defining the cytotoxicity of natural products by the American National Cancer Institute and in scientific literature [51,99,137,138]. Thus, additional studies on EOs should be conducted in the perspective of their pharmaceutical use as antimicrobial agents. Multiple, independent assays may be necessary to confirm experimental outcomes, as results may differ depending on the cell type used. However, as an in vitro proof-of-concept on herbal safety, testing at least one type of mammalian cell model with a standardized methodology is strongly recommended in all studies.

## 8. Conclusions

The present review discusses data obtained from 89 research papers regarding the anti-*Candida* activity of EOs collected from 100 species of Lamiaceae plants. EOs, as well as other plant-derived natural products, represent a main source of new drug molecule today, thus investigations on their biological potential is of clinical relevance. For this reason, studies aimed at evaluating the antimicrobial activity of natural products should be conceived following as much as possible the standardized guidelines of the International Committees (CLSI and EUCAST). Some modifications of protocols may be requested because of the complexity of the tested material, however, these changes should not upset the basics of microbiology.

Most of the studies were screening publications with the intention to identify plants with a potential antimicrobial activity, indeed they tested a wide panel of microorganisms, including other fungi and bacteria, and about 10% of species produce an EO very potent against *Candida* species. The identification of plant species particularly active as anti-*Candida* can represent a starting point for a taxonomic approach aimed at screening closely related species that may potentially contain important chemicals in their EO, an approach that turned sometimes very successful for medicinal plants. On the other hand, the revision of papers has highlighted, in most cases, the insufficiency of adequate toxicological analyses that represent a mandatory requirement to define the safety profile of an EO.

## Figures and Tables

**Figure 1 antibiotics-09-00395-f001:**
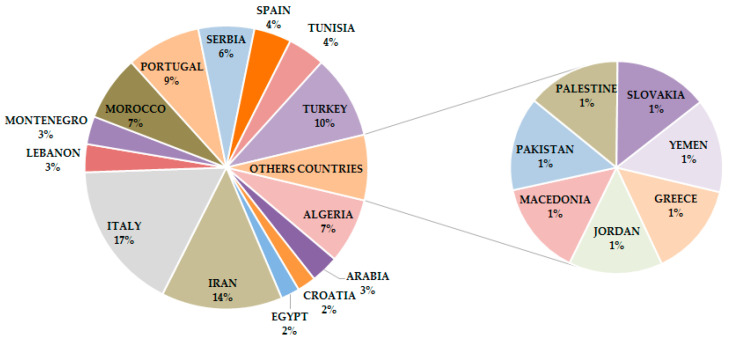
Frequency distribution of reviewed plant species producing EOs with anti-*Candida* activity.

**Figure 2 antibiotics-09-00395-f002:**
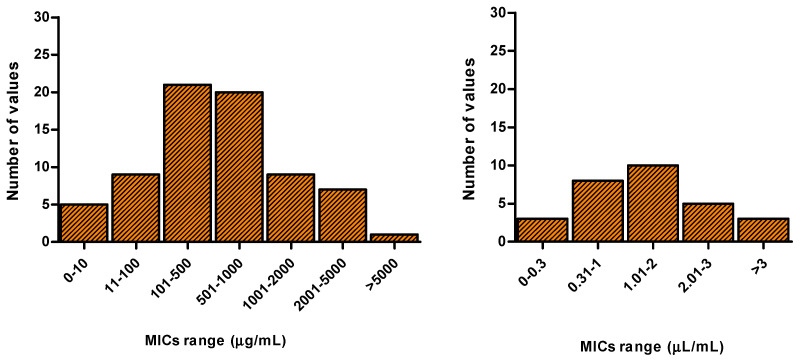
Frequency of distribution of Minimum Inhibitory Concentration (MIC) values related to the selected EOs. Studies reporting MICs obtained from plant species analyzed in different seasons or collected in different regions were excluded from the analysis.

**Figure 3 antibiotics-09-00395-f003:**
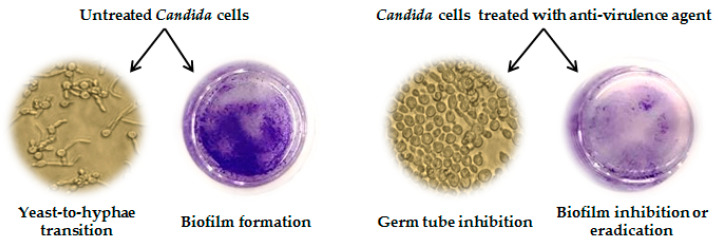
*C. albicans* cultures grown under conditions inducing yeast-to-hyphae transition and in presence of a representative agent inhibiting the expression of virulence factors. Untreated cells undergo transition to filamentous forms and produce a strong biofilm, stained with crystal violet. Treated cells are blastospores and budding cells but not hyphae, and the biofilm is faintly stained.

**Table 1 antibiotics-09-00395-t001:** General overview of the targeted *Candida* spp. and other relevant information of the reviewed research papers.

Ref.	*C.a.* (ref.)	*C.a.* (c.i.)	*C.t.* (ref.)	*C.t.* (c.i.)	*C.gl.* (ref.)	*C.gl.* (c.i.)	*C.gu.* (ref.)	*C.gu.* (c.i.)	*C.k.* (ref.)	*C.k.* (c.i.)	*C.p.* (ref.)	*C.p.* (c.i.)	*C.* spp. (ref.)	*C.* spp. (c.i.)	Anti VF	Others Path.	Cito Tox
[24]	**√**															**√**	
[25]	**√**		**√**					**√**		**√**	**√**						
[26]	**√**																
[27]		**√**														**√**	
[28]	**√**	**√**			**√**	**√**				**√**		**√**		**√**		**√**	
[29]	**√**															**√**	
[30]	**√**	**√**				**√**				**√**							**√**
[31]	**√**															**√**	
[32]	**√**															**√**	
[33]		**√**							**√**		**√**						
[34]	**√**	**√**	**√**					**√**		**√**	**√**				**√**	**√**	**√**
[35]	**√**	**√**	**√**					**√**		**√**	**√**				**√**	**√**	
[36]		**√**														**√**	
[37]	**√**		**√**					**√**		**√**	**√**					**√**	**√**
[38]	**√**	**√**	**√**					**√**		**√**	**√**					**√**	
[39]	**√**				**√**				**√**							**√**	
[40]		**√**														**√**	
[41]	**√**			**√**												**√**	
[42]		**√**														**√**	
[43]	**√**		**√**					**√**		**√**	**√**				**√**	**√**	
[44]					**√**											**√**	**√**
[45]	**√**															**√**	**√**
[46]	**√**	**√**															**√**
[47]														**√**		**√**	
[48]	**√**	**√**			**√**	**√**				**√**		**√**				**√**	
[49]	**√**		**√**		**√**		**√**		**√**		**√**		**√**			**√**	
[50]	**√**															**√**	
[51]	**√**															**√**	
[52]	**√**		**√**		**√**				**√**				**√**			**√**	
[53]	**√**	**√**	**√**			**√**			**√**		**√**			**√**		**√**	
[54]		**√**												**√**			
[55]	**√**															**√**	
[56]	**√**															**√**	**√**
[57]	**√**															**√**	
[58]	**√**		**√**								**√**		**√**	**√**		**√**	
[59]		**√**		**√**								**√**				**√**	
[60]		**√**														**√**	**√**
[61]	**√**	**√**	**√**					**√**		**√**	**√**				**√**	**√**	
[62]	**√**															**√**	
[63]	**√**															**√**	
[64]		**√**														**√**	
[65]	**√**															**√**	**√**
[66]	**√**															**√**	
[67]		**√**														**√**	**√**
[68]	**√**															**√**	
[69]		**√**														**√**	
[70]		**√**								**√**						**√**	
[71]	**√**															**√**	
[72]	**√**															**√**	
[73]	**√**															**√**	
[74]		**√**		**√**												**√**	**√**
[75]	**√**		**√**		**√**											**√**	
[76]	**√**	**√**													**√**		
[77]	**√**	**√**	**√**	**√**	**√**	**√**			**√**	**√**	**√**	**√**	**√**		**√**	**√**	**√**
[78]	**√**		**√**					**√**		**√**	**√**					**√**	
[79]		**√**														**√**	
[80]	**√**	**√**		**√**												**√**	
[81]	**√**			**√**												**√**	**√**
[82]	**√**															**√**	
[83]	**√**	**√**	**√**			**√**			**√**		**√**			**√**		**√**	
[84]	**√**			**√**												**√**	
[85]	**√**															**√**	
[86]	**√**															**√**	
[87]	**√**				**√**				**√**								
[88]	**√**	**√**				**√**		**√**				**√**			**√**		
[89]	**√**															**√**	
[90]		**√**														**√**	
[91]	**√**																
[92]	**√**		**√**					**√**		**√**	**√**				**√**	**√**	**√**
[93]		**√**														**√**	
[94]	**√**															**√**	
[95]	**√**															**√**	
[96]	**√**															**√**	
[97]	**√**															**√**	**√**
[98]	**√**		**√**		**√**				**√**				**√**				
[99]	**√**	**√**													**√**	**√**	**√**
[100]	**√**															**√**	**√**
[101]	**√**															**√**	
[102]		**√**		**√**		**√**				**√**		**√**		**√**			
[103]	**√**															**√**	
[104]	**√**		**√**													**√**	
[105]	**√**															**√**	
[106]	**√**		**√**					**√**		**√**	**√**				**√**	**√**	**√**
[107]	**√**															**√**	
[108]		**√**														**√**	
[109]		**√**														**√**	
[110]		**√**														**√**	
[111]		**√**														**√**	
[112]	**√**	**√**	**√**			**√**		**√**		**√**	**√**				**√**	**√**	

C.a.: *C. albicans*; C.t.: *C. tropicalis*; C.gl.: *C. glabrata*; C.gu: *C. guillermondii*; C.k.: *C. krusei*; C.p.: *C. parapsilosis*; C.spp.: other *Candida species*; ref.: reference strain; c.i.: clinical isolate; anti VF: anti virulence factors; path.: pathogens; citotox.: cytotoxicity. √: indicates targeted strains and/or analysis reported on the reviewed paper. See references for details.

**Table 2 antibiotics-09-00395-t002:** Minimum Inhibitory Concentrations (MICs) of the EOs with anti-*Candida* activity.

Species	Plant part	Method	*Candida* spp.	MIC µg/mL	MIC µL/mL	Ref.
*Calamintha nepeta* (L.) Savi subsp. *nepeta* (Italy)	AP	HD	*C. parapsilosis*		1.25	
			*C. tropicalis*		1.25	[75]
			*C. albicans*		1.25	
*Calamintha nepeta* subsp. *glandulosa* (Req.) P.W.Ball= *Calamintha glandulosa* (Req) Benth.	AP	SD	*C. albicans*	780–12,480		[41]
*Coridothymus capitatus* (L.) Rchb. F.	I	HD	*C. albicans*	128		[70]
	F	HD	*C. glabrata*		1.25	[74]
	AP	HD	*C. albicans*	800		[27]
	F	HD	*C. albicans*		1.25	[74]
*Hymenocrater longiflorus* Benth.	AP	HD	*C. albicans*	240		[26]
*Lavandula angustifolia* Mill.	I	HD	*C. albicans*	512		[70]
*Lavandula luisieri* (Rozeira) Rivas Mart.	AP	HD	*C. albicans*		1.25–2.5	[111]
*Lavandula multifida* L.	AP	HD	*C. tropicalis*		0.32	[110]
			*C. parapsilosis*		0.32	
			*C. albicans*		0.32	
*Lavandula viridis* L’Hér.	AP	HD	*C. parapsilosis*		1.25	
			*C. tropicalis*		1.25–2.5	[109]
*Mentha australis* R.Br.	AP	HD	*C. glabrata*	1.23 *		[61]
			*C. krusei*	1.02 *		
*Mentha mozaffarianii* Jamzad	AP (cultivated)	HD	*C. albicans*	0.39		[102]
	AP (wild)	HD	*C. albicans*	0.78		
*Mentha pulegium* L.	AP	HD	*C. tropicalis*		1.25	[91]
			*C. parapsilosis*		1.25	
			*C. albicans*		1.25	
*Mentha spicata* L.	AP	HD	*C. glabrata*	256		[33]
			*C. tropicalis*		1.25	[91]
			*C. parapsilosis*		1.25	
			*C. albicans*		1.25	
*Mentha suaveolens* Ehrh.	L	HD	*C. albicans*	390		[89]
	com.	HD	*C. albicans*	760		[99]
	AP	HD	*C. albicans*	4		[51]
*Micromeria inodora* (Desf.) Benth.	AP	HD	*C. albicans*	1000		[38]
*Nepeta asterotricha* Rech. F.	PM	HD	*C. albicans*	500–2000		[57]
*Nepeta cataria* L.	AP	HD	*C. tropicalis*		0.125	[108]
			*C. glabrata*		0.25	
			*C. krusei*		0.5	
			*C. albicans*		0.125	
*Nepeta transcaucasica* Grossh.	AP	HD	*C. parapsilosis*	750		[63]
			*C. tropicalis*	375		
*Origanum boissieri* Ietsw.	AP	HD	*C. parapsilosis*	125		[65]
			*C. tropicalis*	250		
			*C. utilis*	125		
			*C. albicans*	125–250		
*Origanum ehrenbergii* Boiss.	AP	HD	*C. albicans*	800		[28]
*Origanum syriacum* L.	I	HD	*C. albicans*	128		[70]
	AP	HD	*C. albicans*	800		[28]
*Origanum vulgare* subsp. *virens* (Hoffmanns. & Link) Ietsw.	AP	HD	*C. tropicalis*		0.32–1.25	[105]
			*C. parapsilosis*		0.64–1.25	
			*C. albicans*		0.32–1.25	
*Phlomis floccosa* D. Don.	AP	HD	*C. albicans*	625		[79]
*Plectranthus barbatus* Andrews	AP	HD	*C. albicans*	550		[80]
	R	SD	*C. albicans*	32–64		[55]
	St	SD	*C. albicans*	32–64		
	L	SD	*C. albicans*	32–64		
*Plectranthus caninus* Roth	R	SD	*C. albicans*	32–64		[55]
	St	SD	*C. albicans*	32–64		
	L	SD	*C. albicans*	32–64		
*Plectranthus cylindraceus* Hochst. ex Benth	AP	HD	*C. albicans*	550		[80]
*Rosmarinus officinalis* L.	L	HD	*C. albicans*	512		[70]
	L	HD	*C. albicans*		0.29	[83]
*Salvia fruticosa* Mill.	L	HD	*C. albicans*	512		[70]
*Salvia mirzayanii* Rech.f. & Esfand.	AP	HD	*C. albicans*	0.32–0.63		[56]
*Salvia x jamensis* J. Compton	AP	HD	*C. albicans*	156		[53]
			*C. glabrata*	156		
			*C. tropicalis*	156		
*Satureja cuneifolia* Ten.	L	HD	*C. albicans*	128		[70]
	AP	HD	*C. albicans*	400		[27]
*Satureja macrosiphon* (Coss.) Maire	F	HD	*C. krusei*		0.25	[81]
			*C. tropicalis*		0.5	
			*C. glabrata*		0.7	
			*C. albicans*		1.5	
*Satureja thymbra* L.	I	HD	*C. albicans*	128		[70]
	AP	SFE	*C. tropicalis*		0.32	[90]
			*C. parapsilosis*		0.32	
	AP	HD	*C. tropicalis*		0.32	
			*C. parapsilosis*		0.32	
	AP	SFE	*C. albicans*		0.32	
	AP	HD	*C. albicans*		0.32	
	AP	HD	*C. albicans*	400		[27]
*Stachys cretica* ssp. *lesbiaca* Rech. Fil.	AP	HD	*C. albicans*	625		[95]
*Stachys cretica* ssp. *trapezuntica* Rech. Fil.	AP	HD	*C. albicans*	625		[95]
*Stachys lavandulifolia* subsp. *lavandulifolia*	AP	HD	*C. albicans*	187.5		[64]
			*C. albicans*	750		
			*C. parapsilosis*	375		
			*C. tropicalis*	93.7		
			*C. krusei*	750		
*Stachys spruneri* Boiss.	AP	HD	*C. albicans*	12.5		[71]
*Thymbra capitata* (L.) Cav.	AP	HD	*C. albicans*		0.32	[87]
*Thymbra spicata* L.	L	HD	*C. albicans*	128		[70]
	AP	HD	*C. albicans*	600		[27]
*Thymus broussonetii* Boiss.	AP	HD	*C. albicans*	250		[93]
			*C. albicans*	450		[66]
			*C. glabrata*	450		
			*C. parapsilosis*	450		
			*C. krusei*	450		
*Thymus catharinae* Camarda	PM	HD	*C. albicans*	250		[47]
*Thymus ciliatus* (Desf.) Benth.	AP	HD	*C. albicans*	430		[66]
			*C. glabrata*	860		
			*C. parapsilosis*	860		
			*C. krusei*	430		
*Thymus herba-barona* Loisel.	AP	HD	*C. tropicalis*		0.32	[112]
			*C. parapsilosis*		0.32	
			*C. albicans*		0.32	
*Thymus hyemalis* Lange	n.a.	HD	*C. albicans*		1.3–2.5	[45]
*Thymus leptobotrys* Murb.	AP	HD	*C. albicans*	230		[66]
			*C. glabrata*	230		
			*C. parapsilosis*	230		
			*C. krusei*	230		
*Thymus maroccanus* Ball	AP	HD	*C. albicans*	250		[93]
			*C. albicans*	460		[66]
			*C. glabrata*	460		
			*C. parapsilosis*	460		
			*C. krusei*	460		
*Thymus pallasicus* Hayek & Velen.	AP	HD	*C. albicans*	500		[107]
			*C. glabrata*	250		
			*C. tropicalis*	500		
			*C. krusei*	250		
			*C. utilis*	250		
*Thymus pallidus* Coss. ex Batt.	AP	HD	*C. albicans*	900		[66]
			*C. glabrata*	900		
			*C. parapsilosis*	900		
			*C. krusei*	900		
*Thymus saturejoides* Coss.	AP	HD	*C. albicans*	890		[66]
			*C. glabrata*	890		
			*C. krusei*	890		
*Thymus syriacus* Boiss.	AP	HD	*C. albicans*	800		[27]
*Thymus vulgaris* L.	L	HD	*C. albicans*		0.312	[24]
	com.	com.	*C. albicans*			[39]
*Thymus willdenowii* Boiss	WP	HD	*C. albicans*	6.9		[84]
	St	HD	*C. albicans*	6.9		
	L	HD	*C. albicans*	13.8		
	I	HD	*C. albicans*	13.8		
*Thymus x viciosoi* (Pau ex R. Morales)	AP	HD	*C. albicans*		0.32	[104]
			*C. tropicalis*		0.32	
			*C. parapsilosis*		0.32	
*Thymus zygis* L. chemotype thymol	n.a.	HD	*C. albicans*		1.3	[45]
*Vitex agnus-castus* L.	I	HD	*C. albicans*	512		[70]
*Zataria multiflora* Boiss	AP	HD	*C. albicans*	250–2000		[94]
			*C. albicans*		0.16	[82]
			*C. tropicalis*	62–500		[94]
*Ziziphora tenuior* L.	AP	HD	*C. albicans*		1.25	[25]
			*C. parapsilosis*		1.25	
			*C. tropicalis*		1.25	

AP: aerial part; com: commercial EO; F: flowers; I: inflorescences; L: leaves; PM: plant material; R: roots; St: stems; com.: commercial; n.a.: not available; HD: hydrodistillation; SD: steam distillation; SFE: solvent-free extraction. *: values expressed as IC_50_ (50% inhibitory concentration)

**Table 3 antibiotics-09-00395-t003:** Plant species producing EOs with anti-virulence activity against *C. albicans* reference strains.

Species	Inhibitory Concentration	Germ Tube Inhibition	Biofilm Inhibition (I) or Disruption (D)	Ref.
*Lavandula luisieri* (Rozeira) Rivas Mart.	at 1/16 MIC	96%	n.d.	[111]
*Lavandula multifida* L.	at 1/8 MIC	66.3%	n.d.	[110]
*Mentha pulegium* L.	at 1/8 MIC	42.3%	n.d.	[91]
*Mentha spicata* L.	at 1/8 MIC	81.8%	n.d.	[91]
*Origanum vulgare* subsp. *virens* (Hoffmanns. & Link) Ietsw.	at 1/8 MIC	88.6%	n.d.	[105]
*Satureja hortensis* L.	at 3 × MIC	n.d.	50% (I)	[97]
*Satureja macrosiphon* (Coss.) Maire	at MIC	n.d.	100% (I)	[81]
*Thymbra capitata* (L.) Cav.	at MIC	n.d.	28.4% (D)	[87]
*Thymus camphoratus* Hoffmanns. & Link	at 1/16 MIC * and MIC ^§^	40%	not specified (I)	[29]
*Thymus carnosus* Boiss	at 1/16 MIC * and MIC ^§^	20%	not specified (I)	[29]
*Thymus vulgaris* L.	at 0.5% (v/v)	100%	n.d.	[39]
*Thymus x viciosoi* (Pau ex R. Morales)	at 1/8 MIC	20%	n.d.	[104]
*Ziziphora tenuior* L.	at 1/16 MIC * and MIC ^§^	80%	not specified (I)	[25]

n.d.: not determined; * MIC values referred to germ tube inhibition; ^§^ MIC values referred to biofilm examinations.

**Table 4 antibiotics-09-00395-t004:** Safety profile of 20 EOs from the Lamiaceae family.

Species	MIC *	Cell viability ^§^	Ref.
*Coridothymus capitatus* (L.) Rchb. F.	625 µg/mL	IC_50_ 127.2 μg/mL ^#^, 138.9 μg/mL	[28]
*Lavandula luisieri* (Rozeira) Rivas Mart.	1.25–2.5 µL/mL	at 0.32 µL/mL: >75%	[111]
*Lavandula stoechas* L.	2.5 µL/mL	at 1.25 µL/mL: <25%	[112]
*Mentha spicata* L.	256 µg/mL (*C. glabrata*)	LD_50_ 279 to 975 µg/mL	[33]
*Mentha suaveolens* Ehrh.	4 µg/mL	IC_50_ 9.15 to 18.20 μg/mL	[51]
	390 µg/mL	IC_50_ >500–1000 μg/mL	[89]
	760 µg/mL	IC_50_ 35.7 μg/mL ^#^, 75.6 μg/mL	[99]
*Ocimum forskolei* Benth.	8600 µg/mL	at 100 µg/mL: 100%	[32]
*Origanum hirtum* Link	1080 µg/mL	IC_50_ 148.5 μg/mL ^#^, 177.1 μg/mL	[99]
*Plectranthus asirensis* J.R.I. Wood	2200 µg/mL	IC_50_ 6.82 to 7.51 μg/mL	[80]
*Plectranthus barbatus* Andrews	550 µg/mL	IC_50_ 4.93 to 4.99 μg/mL	[80]
*Plectranthus cylindraceus* Hochst. ex Benth	550 µg/mL	IC_50_ 3.88 to 3.97 μg/mL	[80]
*Rosmarinus officinalis* L.	1440 µg/mL	IC_50_ >200 μg/mL ^#^	[99]
*Satureja macrosiphon* (Coss.) Maire	1.25 µL/mL	IC_50_ 6.49 μL/mL	[81]
*Stachys cretica* ssp. *lesbiaca* Rech. Fil.	625 µg/mL	at 200 µg/mL: 54%, 77%	[95]
*Stachys cretica* ssp. *trapezuntica* Rech. Fil.	625 µg/mL	at 200 µg/mL: 59%, 67%	[95]
*Teucrium yemense* Deflers	1250 µg/mL	IC_50_ 24.4 µg/mL, 59.9 µg/mL	[32]
*Thymus camphoratus* Hoffmanns. & Link	1110–2230 µg/mL	at 1110 µg/mL: 54% ^#^	[29]
*Thymus carnosus* Boiss	1110 µg/mL	at 1110 µg/mL: 44% ^#^;	[29]
*Thymus herba-barona* Loisel.	0.32 µL/mL	at 0.32 µL/mL: <10%	[112]
*Thymus vulgaris* L.	0.312 µL/mL	IC_50_ 35.4 to >150 µg/mL ^#^	[24]
*Ziziphora tenuior* L.	1.25 µL/mL	at 1.25 µL/mL: <25% ^#^, <50%	[25]

* Minimum inhibitory concentration obtained for *C. albicans* reference strains, unless otherwise specified; ^§^ Cell viability expressed as percentage values are relative to control cells; IC_50_: concentration at which viability was reduced by 50%; LD_50_: concentration at which survival was reduced by 50%. Values are ranges and/or individual values obtained in different cell lines. See references for details; ^#^ Value obtained in cell viability assays using a normal cell line.

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
