# Peer review of "Anti-*Candida* Activity of Essential Oils from Lamiaceae Plants from the Mediterranean Area and the Middle East"

_antibiotics, 2020, doi:10.3390/antibiotics9070395_

Round 1
Reviewer 1 Report
The manuscript contains 139 references and provides a sufficient description of recent studies of Anti-Candida activity of essential oils from Lamiaceae plants. The paper is written clearly and can be interesting for readers from different research areas. Nevertheless, there are some areas for improvement:
1. Moderate English changes required (especially for the section "Anti-Candida activity");
2. Line 128: "The plant species selected, and the composition of essential oils are listed in Table 1." A discussion about the selection should be added to this part. Indeed, it is not clear now - who had selected them? Why these species have been selected?
3. Table 1 contains very valuable data and even can be used as a handbook by readers. But it is really huge - 15 pages! Perhaps it can be moved to the Supplementary section;
4. Figure 1 "Figure 1. Geographical distribution of essential oils of the selected Lamiaceae family plants." What does it mean? It should be reworded to make it clear.
5. Table 2 should be reformatted to make it more useful for readers. For example, the papers' titles (in a short form) can be added to the references;
6. A schematic illustration (infographics) of the virulence factors can make the section "Anti-virulence activity" more informative for readers;
7. The section "Synergistic interaction with commercial antifungal drugs" should be extended. It is too short now;
8. The title of the manuscript is "Anti-Candida activity of essential oils from Lamiaceae plants: a review on mandatory requirements for a standardized evaluation". The authors provide a good review of recent studies in this field, but they say too little about the standardized evaluation. Indeed, it is necessary to cut the title to "Anti-Candida activity of essential oils from Lamiaceae plants" or add a proper discussion about standardisation.
Author Response
REVIEWER 1
1.Moderate English changes required (especially for the section "Anti-Candida activity");
Response. The manuscript has been edited by an English-speaking native person, so we hope it now matches the journal standard.
2. Line 128: "The plant species selected, and the composition of essential oils are listed in Table 1." A discussion about the selection should be added to this part. Indeed, it is not clear now - who had selected them? Why these species have been selected?
Response. The criteria for the selection of plant species have been included (lines 199-203).
3. Table 1 contains very valuable data and even can be used as a handbook by readers. But it is really huge - 15 pages! Perhaps it can be moved to the Supplementary section;
Response. Table 1 has been removed from the main text , and added as Supplementary file (now Table S1).
4. Figure 1 "Figure 1. Geographical distribution of essential oils of the selected Lamiaceae family plants." What does it mean? It should be reworded to make it clear.
Response. According to the suggestion, the legend has been changed in: “Frequency distribution of reviewed plant species producing essential oils with anti-Candida activity”.
5. Table 2 should be reformatted to make it more useful for readers. For example, the papers' titles (in a short form) can be added to the references;
Response. Table 2 has been removed from the main text, and added as Supplementary file (now Table S2). In the revised manuscript, Table 1 details MIC values of the most potent essential oils with anti-Candida activity (<1000 µg/mL and 2 µL/mL). Data are now more useful for readers.
6.A schematic illustration (infographics) of the virulence factors can make the section "Anti-virulence activity" more informative for readers;
Response. Figure 4 has been added to the manuscript. The sketch represents visual results of representative experiments performed to identify anti-virulence agent.
7. The section "Synergistic interaction with commercial antifungal drugs" should be extended. It is too short now;
Response. The section has been slightly modified only in the initial part. Actually, it is shorter than the other sections because only 3 references among the over 100 reviewed examine the essential oils in combination to commercial drugs.
8.The title of the manuscript is "Anti-Candida activity of essential oils from Lamiaceae plants: a review on mandatory requirements for a standardized evaluation". The authors provide a good review of recent studies in this field, but they say too little about the standardized evaluation. Indeed, it is necessary to cut the title to "Anti-Candida activity of essential oils from Lamiaceae plants" or add a proper discussion about standardisation.
Response. The title has been modified in “Anti-Candida activity of essential oils from Lamiaceae plants from the Mediterranean area and the Middle East”.

Reviewer 2 Report
This manuscript is a thorough review of the literature regarding studies of the effects of the Lamiaceae plant family on Candida growth. As the authors point out, it is clear that standardized protocols would improve the ability to interpret and compare data. The authors have analyzed a substantial amount of data and this would be of interest for alternative approaches to anti-Candida treatments.
- SECTION 2. The introduction to section 2 (lines 92-103) is a nice addition to introduce those that are not knowledgeable about plants. Table 1 might be better as a supplemental table.
- SECTION 3. The introductory paragraph is a little confusing. The tenses indicate suggestions for studies/ protocols. This may be what the authors intend but it confuses the reader. Since this is a review and data is being presented - it would be more preferable to state what was done (past tense). Some sentences could be added to state what would have been preferred. Figure 2 would have more impact if the cells were made smaller, similar to how a normal heat map is displayed and preferably fit on one page.
- SECTION 4. Even though there is a substantial amount of data and evidence of effort put in collating the data, it is difficult to identify a major conclusion / conclusions of the effect of oils on Candida. It would perhaps would be more useful if key data from Table 2 was presented in separate tables to emphasize specific points the authors want to make. This could be - most potent plants; diverse results; results by compound etc. Table 2 itself could be included as supplementary data.
- SECTION 5. The results in Table 3 would be more clear if values were clearly presented in a separate column in absence of words. MIC could be presented in a separate column. Instead of writing out of biomass reduction and disruption - this could be included in footnotes.
- SECTION 6. As for Table 3 it is difficult to identify values. Cell lines could be put in separate columns. Avoid including text with values.
- Overall the manuscript requires a proofreading for English. There are areas where phrases are awkward or words misused which reduces the impact of the statements.
- Title: The title could be improved: "review on mandatory requirements for a standardized evaluation" is not quite correct. The focus of the manuscript isn't a review on mandatory requirements; the review is of the effects of oils on Candida. The outcome of the review is a need for a standardized evaluation, and the authors clearly identify this requirement in their manuscript.
Author Response
REVIEWER 2
1.SECTION 2. The introduction to section 2 (lines 92-103) is a nice addition to introduce those that are not knowledgeable about plants. Table 1 might be better as a supplemental table.
Response. Table 1 has been removed from the main text, and added as Supplementary file (now Table S1).
2.SECTION 3. The introductory paragraph is a little confusing. The tenses indicate suggestions for studies/ protocols. This may be what the authors intend but it confuses the reader. Since this is a review and data is being presented - it would be more preferable to state what was done (past tense). Some sentences could be added to state what would have been preferred. Figure 2 would have more impact if the cells were made smaller, similar to how a normal heat map is displayed and preferably fit on one page.
Response. Section 3 has been completely rewritten to better distinguish data presented in the published papers from what could be done to improve microbiological investigations. Figure 2 has been scaled down as much as possible, keeping it readable.
3. SECTION 4. Even though there is a substantial amount of data and evidence of effort put in collating the data, it is difficult to identify a major conclusion / conclusions of the effect of oils on Candida. It would perhaps would be more useful if key data from Table 2 was presented in separate tables to emphasize specific points the authors want to make. This could be - most potent plants; diverse results; results by compound etc. Table 2 itself could be included as supplementary data.
Response. Table 2 has been removed from the main text, and added as Supplementary file (now Table S2). In the revised manuscript, Table 1 details MIC values of the most potent essential oils with anti-Candida activity (<1000 µg/mL and 2 µL/mL). Data are now more useful for readers.
4.SECTION 5. The results in Table 3 would be more clear if values were clearly presented in a separate column in absence of words. MIC could be presented in a separate column. Instead of writing out of biomass reduction and disruption - this could be included in footnotes.
Response. Table 3 (now Table 2) has been modified by adding a separate column (MIC) and footnotes.
5.SECTION 6. As for Table 3 it is difficult to identify values. Cell lines could be put in separate columns. Avoid including text with values.
Response. Maybe reviewer means Table 4? Table 4 (now Table 3) has been modified. Cell lines have been removed from the table but a footnote indicating if a data were achieved on normal of malignant cells has been added. This critical information has been added as well in the text.
6.Overall the manuscript requires a proofreading for English. There are areas where phrases are awkward or words misused which reduces the impact of the statements.
Response. The manuscript has been edited by an English-speaking native person, so we hope it now matches the journal standard.
7. Title: The title could be improved: "review on mandatory requirements for a standardized evaluation" is not quite correct. The focus of the manuscript isn't a review on mandatory requirements; the review is of the effects of oils on Candida. The outcome of the review is a need for a standardized evaluation, and the authors clearly identify this requirement in their manuscript.
Response. The title has been modified in “Anti-Candida activity of essential oils from Lamiaceae plants from the Mediterranean area and the Middle East”.

Round 2
Reviewer 1 Report
The manuscript has been improved. I recommend to accept it.